# The Direct and Spillover Effect of Multi-Dimensional Urbanization on PM_2.5_ Concentrations: A Case Study from the Chengdu-Chongqing Urban Agglomeration in China

**DOI:** 10.3390/ijerph182010609

**Published:** 2021-10-10

**Authors:** Sicheng Wang, Pingjun Sun, Feng Sun, Shengnan Jiang, Zhaomin Zhang, Guoen Wei

**Affiliations:** 1College of Architecture and Urban Planning, Guizhou University, Guiyang 550025, China; scwang@gzu.edu.cn; 2College of Geographical Sciences, Southwest University, Chongqing 400700, China; sunpj031@nenu.edu.cn; 3College of Geography and Ocean Sciences, Nanjing University, Nanjing 210023, China; dg1927028@smail.nju.edu.cn (F.S.); dz1627002@smail.nju.edu.cn (S.J.); 4College of Management, Shenzhen Polytechnic, Shenzhen 518000, China

**Keywords:** PM_2.5_ concentrations, urbanization, spillover effect, spatial regression method, Chengdu-Chongqing urban agglomeration (CUA)

## Abstract

The Chengdu-Chongqing urban agglomeration (CUA) faces considerable air quality concerns, although the situation has improved in the past 15 years. The driving effects of population, land and economic urbanization on PM_2.5_ concentrations in the CUA have largely been overlooked in previous studies. The contributions of natural and socio-economic factors to PM_2.5_ concentrations have been ignored and the spillover effects of multi-dimensional urbanization on PM_2.5_ concentrations have been underestimated. This study explores the spatial dependence and trend evolution of PM_2.5_ concentrations in the CUA at the grid and county level, analyzing the direct and spillover effects of multi-dimensional urbanization on PM_2.5_ concentrations. The results show that the mean PM_2.5_ concentrations in CUA dropped to 48.05 μg/m^3^ at an average annual rate of 4.6% from 2000 to 2015; however, in 2015, there were still 91% of areas exposed to pollution risk (>35 μg/m^3^). The PM_2.5_ concentrations in 92.98% of the area have slowly decreased but are rising in some areas, such as Shimian County, Xuyong County and Gulin County. The PM_2.5_ concentrations in this region presented a spatial dependence pattern of “cold spots in the east and hot spots in the west”. Urbanization was not the only factor contributing to PM_2.5_ concentrations. Commercial trade, building development and atmospheric pressure were found to have significant contributions. The spillover effect of multi-dimensional urbanization was found to be generally stronger than the direct effects and the positive impact of land urbanization on PM_2.5_ concentrations was stronger than population and economic urbanization. The findings provide support for urban agglomerations such as CUA that are still being cultivated to carry out cross-city joint control strategies of PM_2.5_ concentrations, also proving that PM_2.5_ pollution control should not only focus on urban socio-economic development strategies but should be an integration of work optimization in various areas such as population agglomeration, land expansion, economic construction, natural adaptation and socio-economic adjustment.

## 1. Introduction

While China’s rapid urbanization has significantly improved people’s living standards, social well-being and economic development, it has also increased the concerns of PM_2.5_ air pollution caused by industrial production agglomeration, disorderly urban expansion and automobile exhaust emissions [1,2]. Long-term exposure to PM_2.5_ increases the survival risk of residents and causes nearly 1.3 million premature deaths in China every year [3]. The rapid transmission of PM_2.5_ pollution and its components is explicitly linked to human well-being, affecting public health, causing global warming and threatening regional sustainable development [4]. Cities are the primary sources of PM_2.5_. Previous studies exploring the correlation between urbanization and PM_2.5_ concentration have been used to develop and support China’s urban air quality optimization strategies [5,6]. However, given the cross-regional transmission of PM_2.5_ components due to atmospheric circulation, cross-regional trade and the enhanced integration of urbanization activities (e.g., industrial economy, public life and urban expansion), PM_2.5_ pollution is affected not just by local urbanization but also by the spillover effects of urbanization (i.e., SEU, which refers to the external benefits of urbanization on PM_2.5_ concentrations in other regions) activities from neighboring areas [7,8,9].

Previous studies have largely explored the direct effects of urbanization (i.e., DEU) on PM_2.5_ concentrations using the Pearson correlation coefficient and linear regression model, focusing on the spatial dependence of urbanization and PM_2.5_ concentrations at the administrative district level [10,11]. However, the spillover effects of urbanization and the spatial agglomeration pattern of PM_2.5_ concentrations at the grid level have generally been overlooked [12]. Spatial regression models, such as the spatial error model (SEM), spatial lag model (SLM) and spatial Durbin model (SDM), can realize the integration of endogenous and exogenous nature of the driving effects and identify the direct and spillover effects of urbanization on PM_2.5_ spatial distribution [13]. Moreover, in the study of the spatial distribution pattern of PM_2.5_ concentrations, integrating the analysis objects at the grid and administrative unit levels would be extremely useful in the decision-making reference value and refined expressions.

Some studies have briefly discussed the impact of different types of urbanization (e.g., population, economic and land urbanization) on air quality and the atmospheric environment [14,15]. With reforms in the household registration system and the rise of small cities, China’s population urbanization (PU) level has rapidly increased, resulting in the widespread use of kitchen fumes and cars and generating large amounts of PM_2.5_ pollutants in daily life [16,17]. Rapid economic urbanization (EU) has boosted urban populations and the economic carrying capacities of central cities but has also caused high PM_2.5_ emissions from industrial production and energy consumption of large diesel engines [18]. Studies have also shown the impact of urban commercial cooperation and transnational trade on increasing and spreading PM_2.5_ pollution levels [19]. In addition, the rapid rise in urban residents, high building density and increased impervious surface coverage (i.e., land urbanization or spatial urbanization) in China’s cities have generated large amounts of road, construction and storage yard dusts and have intensified urban heat island effects with affecting the settlement and diffusion of fine particles [20,21,22]. These studies highlight that different dimension of urbanization can have differentiated effects on PM_2.5_ concentrations.

However, most of the studies have focused on a specific aspect of urbanization. The comprehensive and systematic assessment of urbanization and driving mechanism analysis have been largely overlooked [23]. Realizing the transformation of driving analysis objects from single-dimensional to multi-dimensional urbanization is crucial in understanding the complex driving mechanisms of urbanization on PM_2.5_ agglomeration and diffusion.

As a highly integrated city cluster, urban agglomerations have become an important form of urban development in China. The Yangtze River Delta urban agglomeration, the Beijing-Tianjin-Hebei urban agglomeration and the Guangdong-Hong Kong-Macao Greater Bay Area have become important urban clusters in the world, with concentrated populations, highly developed economies and complete urban structures [24]. However, air pollution levels in these urban agglomeration areas are higher and more dangerous than in other regions. For example, China’s Air Pollution Prevention and Control Action Plan (2013–2017) stated that the Beijing-Tianjin-Hebei, Yangtze River Delta, Pearl River Delta and Chengdu-Chongqing urban agglomeration (CUA) need to implement special emission limits for air pollutants to control regional air quality [25]. The CUA is an emerging urban agglomeration approved by the State Council of China in 2016. It is located in the strategic hub area where China’s “Belt and Road Initiative” and the Yangtze River Economic Belt meet (Figure 1) [26]. According to the National Bureau of Statistics of China, the urbanization levels of Chongqing and Chengdu in 2015 were 62.6% and 70.6% and the average PM_2.5_ concentrations in the main urban areas were 54.0 μg/m^3^ and 64.2 μg/m^3^, respectively. More than ten air pollution incidents have been recorded in this area in recent years [27]. At present, there are obvious research gaps on the PM_2.5_ driving mechanism of China’s emerging urban agglomerations, including in the CUA, although these urban agglomerations are essential for China to achieve economic growth and urbanization transformation in the future [28].

Taking CUA as the research area, the objectives of this study are as follows: (1) to identify the spatial autocorrelation and spatio-temporal trend evolution of PM_2.5_ concentrations in the CUA; (2) estimate the driving influence of urbanization, natural factors, socio-economic factors and other control variables on the spatial distribution of PM_2.5_ concentrations; and (3) compare the direct and spillover effects of multi-dimensional urbanization on PM_2.5_ concentrations. This study aims to explore the driving mechanism of PM_2.5_ concentrations in the CUA given its rapid urbanization and evaluates the influence of the cross-regional urbanization activities on PM_2.5_ concentrations. The results can provide support for coordinating the relationships between urban construction, air quality and the sustainable development of the CUA.

## 2. Study Area and Materials

### 2.1. Study Area

The CUA is centered on Chongqing and Chengdu and encompasses the cities of Zigong, Luzhou, Deyang, Mianyang, Suining, Neijiang, Leshan, Nanchong, Meishan, Yibin, Guang’an, Dazhou, Ya’an and Ziyang [29]. To ensure a sufficient sample size in the spatial regression model, the regression analysis was carried out at the county level, containing a total of 142 sample units (Figure 1). Considering the data availability, the study period was set to 2000–2015; all spatial data projections were set to Krasovsky 1940 Albers and extracted into 142 administrative regions based on Zonal Statistics tool of ArcGIS (Table 1).

### 2.2. PM_2.5_ Concentrations

The Atmospheric Composition Analysis Organization (ACAG) provided the corrected PM_2.5_ concentrations (V4.GL.03, 0.05° × 0.05°, contains “all ingredients”) of the CUA satellite-derived geographic weighted regression (GWR) from 2000 to 2015 (http://fizz.phys.dal.ca/~atmos/martin/?page_id=1751 (accessed on 7 June 2021)) [30]. Python 2.7 (http://www.python.org (accessed on 20 June 2021)) was used for vector cropping and in obtaining the annual average series PM_2.5_ concentrations for all pixels.

### 2.3. Urbanization

Based on recommendations from existing research, this study selected three indicators (i.e., population, land and economic urbanization) to characterize the level of urbanization in the CUA [14]. Population urbanization is a basic aspect of urbanization and is often defined by the proportion of the non-agricultural population, reflecting the changes in urban spatial structure caused by population migration from rural into urban areas. In this study, the permanent urban population percentage was used to characterize the level of population urbanization. The data were obtained from the National Bureau of Statistics of China, “*Sichuan Statistical Yearbook*” and the “*Chongqing Statistical Yearbook*” 2000–2015 [31]. Land urbanization is the process of transforming land-use attributes from agricultural land to urban construction land. It is often accompanied by the differentiation of urban land structures and the fragmentation of green spaces and is usually indicated by the proportion of built-up area [12]. The land-use dataset was provided by the Resources and Environmental Sciences, Chinese Academy of Sciences and was based on Landsat8 remote sensing images. Using visual interpretation, six primary land-use types (i.e., farmland, woodland, grassland, waters, residential land and unused land) and 25 secondary types (https://www.resdc.cn/ (accessed on 10 June 2021)) were generated and used in this study.

Urbanization is not limited to the basic forms, such as population agglomeration and land expansion, but also involves changes in the economic structure, such as industrial transformation and scale production. We measured economic urbanization using GDP density [32]. Previous urbanization studies have supported the use of GDP density as a proxy for non-agricultural economic data, indicating the intensity of socio-economic activities in urbanization [33]. The spatialized dataset was generated based on the GDP statistics at the county level and incorporated land-use types, nighttime light intensity and the density of residential areas closely related to human economic activities [34]. The GDP density grid data were obtained from the Resources and Environmental Sciences, Chinese Academy of Sciences (2000, 2015) (1 km × 1 km), with a data unit of 10,000 yuan/km^2^ (https://www.resdc.cn/ (accessed on 11 June 2021)).

### 2.4. Control Variables

Aside from urbanization, the accumulation and dispersion of PM_2.5_ are affected by factors, such as land-use types, meteorological conditions and degree of forest coverage [14,35]. For example, Duan et al. [36] found that the PM_2.5_ concentrations of Lushan Mountain in China vary due to the altitude and slope. Cheng et al. [37] confirmed that the average temperature, air pressure and relative humidity have significant negative effects on PM_2.5_ concentrations in Lanzhou, China. Zheng et al. [38] concluded that precipitation is one of the main elements to remove aerosol particles in the atmosphere.

In order to improve the fitting effect and experimental scientificity of the spatial regression model, we selected the elevation (*dem*), slope (*slo*), average annual temperature (*tem*), average relative humidity (*hum*), average air pressure (*ap*), average wind speed (*wind*), normalized vegetation coverage index (*ndvi*) and average annual precipitation (*pre*) as natural control variables. The altitude and slope data were derived from the GDEMV2 DEM of the Computer Network Information Center of the Chinese Academy of Sciences, with 30 m spatial resolution (2000, 2015) (http://www.gscloud.cn (accessed on 15 June 2021)). The spatial distribution data for the temperature, relative humidity, air pressure, wind speed and precipitation were derived from the meteorological spatial dataset provided by the National Earth System Science Data Center of China (2000, 2015) (https://www.resdc.cn/ (accessed on 15 June 2021)). The NDVI spatial distribution data were derived from MOD13A3, with a spatial resolution of 1 km (2000, 2015) (https://ladsweb.modaps.eosdis.nasa.gov/search/order/1/MOD13A3--6 (accessed on 16 June 2021)).

Some socio-economic variables have also been found to impact PM_2.5_ concentrations. For example, Wang et al. [39] concluded that air pollutant emissions are related to the goods and services consumed, as the production of goods often requires inputs of raw materials and energy forms produced by heavy industry and the spread of pollutant components is accelerated through cross-regional trade. Qian et al. [40] found that increased building density affects the urban meteorological environment, leading to rapid precipitation and condensation of urban air pollutants. Andrea et al. [41] found that by reducing agricultural pollutant emissions, especially the use of chemical fertilizers, a relative reduction in PM_2.5_ can be achieved. We selected per-capita retail sales of consumer goods (*rscg*), per-capita real estate investment (*rei*) and agricultural fertilizer application (*afa*) to measure the impact of socio-economic factors, such as commercial trade, the building development intensity and agricultural operation intensity, on PM_2.5_. These data were obtained mainly from the “*Sichuan Statistical Yearbook*” and the “*Chongqing Statistical Yearbook*” on the statistics of socio-economic indicators at the county level.

## 3. Methods

Figure 2 shows the technical framework of this research and outlines the research structure, data sources and methods of this study.

### 3.1. Spatial Dependence Pattern and Spatio-temporal Trend Analysis

The spatial dependence pattern and evolution trend of PM_2.5_ concentrations in the CUA were analyzed at the grid and county administrative levels. The global autocorrelation Moran’s *I*, calculated using ArcGIS 10.6 (produced by Environmental Systems Research Institute, RedLands, US), was used to analyze the spatial agglomeration of PM_2.5_ concentrations in urban agglomerations [42]. The hot spot and cold spot analyses were used to identify the hot spot and cold spot distributions of PM_2.5_ concentrations in local regions [43]. The global Moran’s *I* calculation formula is as follows:(1)$$I=\frac{n\sum_{i=1}^{n}\sum_{j\neq 1}^{n}Wij(xi-\bar{x})(xj-\bar{x})}{\sum_{i=1}^{n}\sum_{j=1}^{n}Wij\sum_{i=1}^{n}{\left(xi-\bar{x}\right)}^{2}}$$
where *x_i_* and *x_j_* are the PM_2.5_ concentrations values of the *i* and *j* units and *W_ij_* is the spatial weight matrix. The values of *I* always fall within the [−1, +1]; the greater the absolute value, the stronger the correlation. Moran’s *I* values significantly above zero indicate positive spatial dependence, values significantly below zero indicate negative spatial dependence and zero means no agglomeration. In addition, we used the hot spot analysis tool (Getis-OrdGi* index) in ArcGIS to measure the high/low clustering mode of PM_2.5_ concentrations using the expression:(2)$$Gi^{*}(d)=\sum_{i=1}^{n}Wij(d)xi/\sum_{i=1}^{n}xj$$

The spatial relations and spatial weights were determined based on Fixed Distance and Euclidean distance methods based on the suitability comparison of the results by using different methods. When *Gi*(d)* exceeds zero, the observation unit belongs to a hot spot area—that is, the concentrated area with high PM_2.5_ concentrations. When the value is less than zero, the observation unit belongs to a cold spot area—that is, the concentrated area with low PM_2.5_ concentrations.

In addition, the Theil–Sen trend method was used to measure the spatio-temporal variation trends of the PM_2.5_ concentrations in the CUA [44]. This method was implemented in Python, a robust non-parametric statistical algorithm with strong resistance to measurement errors. The calculation formula is as follows:(3)$$S_{PM2.5}=Median\left(\frac{x_{i}-x_{j}}{i-j}\right),2000\leq i<j\leq 2015$$
where *S* is the median slope of the *n*(*n*−1)/2 data combinations and *x_i_* and *x_j_* are the PM_2.5_ concentration values of a grid in years *i* and *j*. Values significantly greater than zero indicate increasing PM_2.5_ concentrations, while values significantly less than zero suggest declining PM_2.5_ concentrations.

### 3.2. Spatial Regression Model

Given the spatial dependence of PM_2.5_ concentrations, spatial effects were incorporated into the regression simulation and the spatial regression models were established in place of ordinary least squares (OLS) regression. Spatial regression models can show the interaction effects of endogenous and exogenous variables [45]. The first were the endogenous interactions between dependent variables, which could be described as the PM_2.5_ pollution in neighboring areas that caused the change of PM_2.5_ concentrations in a specific location through the migration effect and reflect the pollution externality of PM_2.5_ contaminants through the spatial interaction mechanism. The SEM can be used to identify the error impact of dependent variables [46]. The second were the exogenous interaction effects between independent variables, which could be described as the spatial spillover effect of independent variables. It means that independent explanatory variables in a specific location (e.g., urban activity and natural source factors) have external benefits and may impact PM_2.5_ concentrations in neighboring locations, consistent with the cross-regional effects of factors, such as urbanization [47]. The SLM can be used to reveal the spatial spillover effect of independent variables.

The SDM integrates the interpretation function of the SEM and SLM, allowing for the assessment of internal spatial dependence of the dependent variable and the external spatial dependence of the independent variable (the calculation formulas of SDM and SLM are not explained in detail, please refer to the study of Du et al. [48]. In this study, the SDM was used to analyze the impact factors of PM_2.5_ concentration and the influence of multi-dimensional urbanization on PM_2.5_ concentrations. The formula is as follows:(4)$$Y=\rho WY+X\beta +WX\theta +\alpha K_{n}+\varepsilon$$
where *Y* is a dependent variable of order *n* ∗ 1; *X* is an explanatory variable of order *n* ∗ *k*, including key explanatory variables (urbanization) and control variables (natural and socio-economic variables); *ρ*, *α*, *β* and *θ* are parameters to be estimated; *ε* is a normally distributed disturbance term with a diagonal covariance matrix; *W* is the spatial weight matrix reflecting the location relationship of the region; *WY* is the spatial lag dependent variable; and *WX* is the spatial lag independent variable.

To improve the rationality of model fitting, the following aspects were considered: (1) urbanization and the control variables are assumed to be the driving factors of PM_2.5_ concentrations in the CUA. This study focused the analysis on urbanization. (2) Using SPSS23.0 (produced by IBM SPSS Statistics, Amunk, NY, USA), the variables were tested for multicollinearity. The results showed that the Condition Index and Variance Inflation Factor (VIF) of *tem*, *hum*, *afa* and *pre* were all over the threshold and were therefore eliminated. (3) All variables were normalized to eliminate heteroscedasticity. (4) The optimal model selection principle was established. Based on the research of Du et al. [48], after the Lagrangian multiplier (LM) and residual spatial autocorrelation passed the significance test, the likelihood ratio (LR) test was used to determine whether SDM can be reduced to SLM and SEM. If *ρβ* + *θ* = 0 passes the significance test, SDM can be simplified to SEM. If *θ* = 0 passes the significance test, SDM can be simplified to SLM. If both are rejected, SDM is most suitable for the study’s fitting analysis. Spatial regression analysis was implemented in the MATLAB (produced by MathWorks, Natick, MA, USA) Spatial Regression Toolbox.

## 4. Results

### 4.1. Spatial Dependent Pattern and Evolution Trend of the PM_2.5_ Concentrations

Figure 3 shows the spatial distribution pattern of PM_2.5_ concentrations in the CUA from 2000 to 2015. The overall distribution pattern of PM_2.5_ concentrations in the CUA was relatively stable. In 2000, the PM_2.5_ concentration was relatively high in Northwest China and was relatively low in the southeast. From 2005–2015, cities such as Jiang, Zigong and Meishan, formed the core of high concentration value areas, forming a distribution pattern with a high concentration in the southwest and a low concentration in the southeast. The overall PM_2.5_ concentrations in the CUA have decreased significantly. Moreover, in 2000 and 2015, the average PM_2.5_ concentrations in the grid layer were 102.06 μg/m^3^ and 48.05 μg/m^3^, respectively, equivalent to a mean annual decrease of 4.6%.

For the given study period, the CUA had a large area with PM_2.5_ concentrations exceeding 35 μg/m^3^. According to the World Health Organization (WHO) Air Quality Guidelines (2005), long-term exposure to environments with PM_2.5_ concentrations exceeding 35 μg/m^3^ have considerably higher mortality risk than those with less than 10 μg/m^3^. In 2000, more than 99% of the CUA had concentrations of more than 35 μg/m^3^, while the proportion decreased to 91% in 2015 (Figure 4). The proportion was still very high, indicating that air pollution was still a major problem in this area.

The Theil–Sen trend analysis shows that most in the CUA experienced a convergence of PM_2.5_ concentrations from 2000 to 2015 (Figure 5). For the given study period, 92.98% of the areas experienced a slowly descend in PM_2.5_ concentrations, particularly in Suining, Leshan, Ya’an, Mianyang and Dazhou. The PM_2.5_ concentrations in 5.66% of the agglomeration had sharply declined, primarily in the main urban area of Chengdu, eastern Meishan, northern Leshan, Ziyang, Guang’an and the main urban area of Chongqing. In 1.36% of the agglomeration, the PM_2.5_ concentrations had slowly increased, mainly in Shimian, Xuyong and Gulin counties.

To evaluate the spatial dependence pattern of PM_2.5_ concentrations in the CUA, the overall Moran’s *I* from 2000 to 2015 was calculated at the county administrative level (Figure 6). The results show that the Moran’s *I* of the PM_2.5_ concentrations were all greater than zero, with an average value of 0.578. This suggests that the spatial distribution of PM_2.5_ concentrations in the CUA has a significant positive spatial dependence (z-scores in each year are more than 2.58 and p-values are less than 0.01) and that high-polluting cities are often adjacent. Figure 7 shows the distribution of cold and hot spots of PM_2.5_ concentrations in the CUA in 2000 and 2015. Consistent with the changes in the spatial distribution, the spatial heterogeneity of cold and hot spots of PM_2.5_ concentrations was significant from 2000 to 2015, presenting a spatial dependence pattern of “cold spots in the east and hot spots in the west”. In 2000, Hanyuan-Anyue-Langzhong was largely used as the node forming a “reverse-C” shape of cold/hot spot boundaries. The hot spots were concentrated in highly polluted areas, such as Chengdu, Deyang, Suining, Meishan and Mianyang, in the western part of the Chengdu-Chongqing urban agglomeration. In 2015, the hot and cold dividing boundary was transformed into a “ring”-shaped structure with Ebian, Mianzhu, Nanchuan and Gulin as the nodes. The center of the hot spot moved south and the cold spot only included Dazhou, Mianyang, Nanchong and the northern part of Chongqing (such as Wanchuan, Yunyang, Liangping) after shrinking. From 2000 to 2015, a large number of cold spot cities changed to non-significant areas and their proportion dropped from 29.57% to 19.72%, while hot spot areas remained above 30% (Figure 7).

### 4.2. Driving Impact of Urbanization on PM_2.5_ Concentrations

The PM_2.5_ concentrations showed significant spatial dependence; therefore, the spatial effects have to be considered when measuring the driving factors of PM_2.5_ concentrations. Meanwhile, in the trend analysis, the PM_2.5_ concentrations in the main urban areas of Chengdu and Chongqing decreased significantly, which arouses our attention to the driving influence of urbanization on the spatial distribution of PM_2.5_ concentrations. To explore the impact of urbanization on the spatial distribution of PM_2.5_ concentrations, we used multi-dimensional urbanization as the key explanatory variable and the natural and socio-economic parameters as control variables in generating the spatial regression model. To demonstrate the spatial regression optimal model selection process, take the population urbanization as an example.

The first step is to confirm whether spatial effects should be introduced into the regression model. Both LM-SLM and LM-SEM rejected the null hypothesis, indicating no spatial lag term and spatial error term at the 1% confidence level. The residual space autocorrelation test shows that the residual Moran’s *I* of spatial regression is closer to zero than OLS. Both tests proved the necessity that spatial regression models are introduced. Then, when the substitutability of the SLM and SEM models were tested using the LR, we found that both the LR-SLM and the LR-SEM were statistically significant at the 1% confidence level. This means that using SDM in the regression analysis can better reveal the driving mechanism of PM_2.5_ concentrations than SLM or SEM and simultaneously characterize the interaction of variable endogeneity and exogeneity and the differential impact of urbanization on PM_2.5_ concentrations in local and neighboring areas. Finally, the results of the SDM were analyzed.

Table 2 shows the results of the SDM calculation for population urbanization and comprehensive urbanization (*lncu*) for 2000 and 2015. Among them, referring to the study of Du et al. [12], the comprehensive urbanization index (the urbanization of each dimension was standardized, added and averaged and was again standardized) was introduced into the SDM to reflect the impact of the comprehensive urbanization level on the PM_2.5_ concentrations. Using population urbanization as an example, the results show that the coefficients of *lnpop* and *W*lnpop* for 2000 and 2015 were significantly greater than zero and showed an increasing trend (Table 2, columns 2 and 3). This indicates that the increase in population urbanization has a significant positive impact on the spatial distribution of PM_2.5_ concentration. This finding is also applicable to comprehensive urbanization (Table 2, columns 5 and 6).

In addition, the comparison of regression coefficients showed (Table 2, columns 2 and 3) that population agglomeration was not the only gain factor for the PM_2.5_ concentration growth in the CUA. The control variables, such as the slope, commercial trade, air pressure and building development intensity, were all positive driving factors that caused the CUA’s relatively high concentrations in 2000, with regression coefficients of 0.075 (*lnslo*), 0.018 (*lnrscg*), 0.797 (*lnap*) and 0.207 (*lnrei*). NDVI and elevation had a significant impact on preventing the increase of PM_2.5_ concentrations in the region, especially in 2015 when the regression coefficients were –0.050 (*lnndvi*) and –0.624 (*lndem*). The significant decrease of PM_2.5_ concentrations in the CUA in 2015 (Figure 3) may be related to the rapid convergence of the driving influence of *lnap* and *lndem* on PM_2.5_. The gain effect of average air pressure decreased by 37.892% compared to 2000, while the suppression effect of elevation increased by 345.714%.

### 4.3. Difference between the Direct and Spillover Effect of Multi-Dimensional Urbanization

Table 2 shows the difference in the regression coefficient between the weighted variables and the original variables, verifying that urbanization under the spatial effect has a differential impact on PM_2.5_ concentrations of local and neighboring areas. To explore the differential effects of urbanization in local and neighboring areas, we constructed SDM models using multi-dimensional urbanization and the PM_2.5_ concentration as independent and dependent variables, analyzing whether the population, land and economic urbanization have the same impact on PM_2.5_ concentrations and its direct and spillover effects. Table 3 lists the direct and spillover impact of multi-dimensional urbanization for 2000 and 2015.

We found that the impact of the direct and spillover effect of urbanization on PM_2.5_ concentrations was generally: land urbanization > economic urbanization > population urbanization. The impact of multi-dimensional urbanization on PM_2.5_ concentrations was declining both locally and in neighboring areas. This shows that urban land expansion has greater contributions to the PM_2.5_ concentration than economic construction and urban–rural population conversion. The general decline in the influence of urbanization indicates that the driving factors of PM_2.5_ distribution may be more diversified.

In terms of population urbanization, the direct effect was not significant in 2000. The elastic coefficient of *lnpop* was 0.083 in 2015, significant at a 1% confidence level. The spillover effect was statistically significant for the two periods with elastic coefficients of 2.266 and 1.370, equivalent to a decrease of 39.54%. In terms of land urbanization, with a confidence level of 1%, the elastic coefficients of the direct effect and spillover effect of *lnland* from 2000 to 2015 have converged, decreasing by 5% and 65.54%, but the direct effect remains the first for multi-dimensional urbanization. In terms of economic urbanization, the elastic coefficients of the direct effects and spillover effects for *lngdp* from 2000 to 2015 have also converged, decreasing by 10.6% and 8.36%. Moreover, we found that the impact of population, land, economic and comprehensive urbanization on the PM_2.5_ concentrations in neighboring cities was stronger than local urbanization, which means that the spillover effect of urbanization is significantly stronger than the direct effect. For example, in 2000 and 2015, the elastic coefficients of the direct effect of comprehensive urbanization on PM_2.5_ concentrations were 0.169 and 0.155 (1% confidence level), which were only 2.31% and 7.21% of the spillover effect in those periods.

## 5. Discussion

### 5.1. Explanation of the Driving Influence of Urbanization on PM_2.5_ Concentrations

Why was the influence of urbanization on the PM_2.5_ concentration in 2000–2015 not pronounced? As shown in Table 2, the population urbanization and comprehensive urbanization of the region are significantly weaker than the gain contributions of the control variables, such as the slope, air pressure, commercial trade and building development intensity to the PM_2.5_ concentrations. There are two possible reasons for this phenomenon. First, PM_2.5_ concentrations result from multiple factors, including social, economic and ecological factors, not just due to urbanization-related human activities [49]. For example, CUA areas are mostly basins with low slopes. Studies have confirmed that strong downdrafts are formed over the basins in this area. The temperature inversion near the ground and the de-ground temperature inversion effect formed after the high airflow sinks inhibit the upward diffusion of PM_2.5_ pollutants and cause the slope to have a strong positive impact on PM_2.5_ concentrations [50].

The second is the impact of the spatial difference arising from the layout center of multidimensional urbanization and PM_2.5_ concentrations. Figure 8 shows the spatial distribution of population, land and economic urbanization. Comparing Figure 3 and Figure 8, we found that the high-value urbanization centers represented by the main urban areas of Chongqing (Yuzhong District, Nan’an District, Yubei District, Jiangbei District) have relatively low PM_2.5_ concentrations. The difference in spatial distribution between urbanization centers and pollution centers weakens the explanatory force of the multi-dimensional urbanization in the regression fitting.

### 5.2. Differential Impact of Multi-Dimensional Urbanization on PM_2.5_ Concentrations

The direct and spillover effects of multi-dimensional urbanization on PM_2.5_ concentrations significantly vary. In terms of population urbanization, the elastic coefficient of DEU was not significant in 2000 and in 2015, it was ranked last among the urbanization aspects (Table 3, row 1). This may be related to the inconsistency of the spatial distribution of population urbanization and PM_2.5_ concentrations, which weakens the direct influence of population urbanization. Comparing Figure 3 and Figure 8, the high-level population urbanization regions, such as Ziliujing District, Dadukou District, Jiangbei District and Nan’an District, have not formed high-value PM_2.5_ agglomeration and their concentrations are lower than the mean value of CUA in 2000 (102.06 μg/m^3^) and 2015 (48.05 μg/m^3^). This observation has also been adapted to the current situation of population urbanization policy and green industrial development of CUA [51]. Taking 2015 as an example, the population urbanization level of Chongqing’s main urban areas, such as Yuzhong District (100%), Dadukou District (97.2%) and Jiangbei District (95.3%), ranked at the top. The construction of large-scale settlements and new urban areas has accelerated urban population agglomeration. Under the new urbanization policies, these regions adhere to green, low-carbon and ecological development, strengthen energy conservation, promote emission reduction and implement eco-environmental protection measures. Taking NO_2_ emissions and smoke in the main urban area of Chongqing as an example, both values decreased by 60.62% and 94.14% from 2000 to 2015 (data from the *Chongqing Statistical Yearbook*). Meanwhile, the radiation capacity of advanced manufacturing and modern service industries in the main urban area of Chongqing has rapidly increased and green industrialization has accelerated. The rapid agglomeration of the urban population, improvements in resident lifestyles and the significant weakening of the ecological predatory effect due to modern green industry have considerably reduced the air pollution effect of population urbanization in the region [52,53].

The SEU elasticity coefficient of population urbanization was significant. We argue that the fact that the cross-regional movement of people under the complex urban system has strengthened the transmission effect of PM_2.5_ concentrations can explain this phenomenon (Table 3, row 1). Lin et al. [54] and Shen et al. [55] concluded that the construction of urban systems, coupled with population mobility and the agglomeration of rural populations into cities, has changed the geographical distribution and scale of pollutants, resulting in considerable regional air quality problems. According to the *Chengdu Statistical Yearbook* and *Chongqing Statistical Yearbook*, from 2000 to 2015, the permanent residents of Chengdu and Chongqing increased by 4.582 million and 8.245 million and the growth rates were 75.55% and 81.32%, respectively. The agglomeration and flow of the urban population increased domestic pollution, resulting in higher population-related PM_2.5_ emissions in neighboring areas.

The elastic coefficients of DEU and SEU of land urbanization are generally at the top of multi-dimensional urbanization under statistical significance. This may be related to the regional land expansion demand and the urban heat island effect [56,57]. Due to national strategic support (e.g., China’s western development strategy, the Yangtze River Economic Belt and CUA), the CUA has expanded considerably to satisfy increased residential housing demand [58]. According to the National Bureau of Statistics of China and Figure 8, the average value of per capita real estate investment in fixed assets in the CUA increased from CNY 476.09 in 2000 to CNY 7272.32 in 2015, with an average annual growth rate of 19.93%. Similarly, the average level of land urbanization increased from 4.94% to 8.94%. Together with atmospheric circulation and urban wind, the increased building construction and road dust exacerbated the air pollution of the local city and caused gain effects on the PM_2.5_ concentrations of neighboring towns [59]. The rapid increase in building density and impermeable surfaces further constricted urban green spaces and vegetation coverage, affecting the adsorption effect on fine particles and influencing the accumulation and diffusion of pollutants through the heat island effect [60,61].

Although the direct and spillover effects of economic urbanization have converged, the elastic coefficient remains relatively high. The direct effect shows that economic urbanization is an important indicator affecting PM_2.5_ concentrations. For the given research period, the average level of economic urbanization in the CUA rose from 6,203,500 yuan/m^2^ to 67,685,900 yuan/m^2^ (Figure 8). The rapid development of the urban economy was accompanied by the rise in industrial production, which has brought about substantial dust pollution and aggravated the local PM_2.5_ concentrations through various complex mechanisms [62,63]. The industry structure of the CUA, led by the secondary industry and its role as a demonstration zone for industrial transfer in eastern China, has promoted greater pressure on energy conservation, emission reduction and environmental governance in the region [64]. The spillover effects may be related to improved regional transportation economy and developments in cross-regional trade [65]. As the connecting point of “the Belt and Road initiative” and the Yangtze River Economic Belt, the CUA is the fourth key area in China’s comprehensive transportation network [66]. For the study period, the region integrated airports, highways, railways and port container transportation, gradually building a transportation network that radiates domestic and foreign regions and the increased local traffic pollution intensified PM_2.5_ levels in neighboring cities [67]. Similarly, based on the transportation system, trade connection between CUA and domestic and foreign markets has strengthened rapidly, causing accelerated transmission of pollutant components under the trade globalization, which further intensify the risk of air pollution in neighboring areas [19].

In conclusion, land urbanization is the best indicator reflecting the impact of multi-dimensional urbanization on PM_2.5_ concentrations, which is particularly evident in its spillover effects. We argue that the possible reason is that land is the spatial carrier of urban economic development and population agglomeration. Given the policies of the Chinese government promoting the construction of national central cities (e.g., Chengdu, Chongqing), metropolitan areas (e.g., Chengdu metropolitan area, Chongqing metropolitan area) and urban agglomerations, land expansion and the transformation of land use properties in the CUA have accelerated, further fragmenting urban green spaces and threatening environmental habitats [68,69,70]. With the support of China’s special institutional environment and rapid urbanization, land urbanization has a more prominent impact on PM_2.5_ concentrations.

### 5.3. Policy Implications, Limitations and Applicability

Although the PM_2.5_ concentrations in the CUA have significantly declined from 2000 to 2015, the impact of urbanization on air pollution should not be underestimated, particularly its spillover effects in neighboring areas. The relationship between urbanization and air quality should be carefully coordinated, focusing on the spillover effects of urbanization on air quality. More work should be carried out to promote the construction of green cities and improvements in air quality. First, policymakers and environmental protection organizations must consider the complex relationship between urbanization and urban ecosystems from the system theory perspective and the human–land coordination concept. Urban development must be optimized in terms of population agglomeration, land expansion and economic construction. For example, emission monitoring of heavy industries should be improved and a system that monitors dust and air quality on construction sites should be developed. The creation and protection of green spaces should be incorporated in new construction works and the construction of pollution prevention and control, treatment and disposal facilities should be integrated into urban planning. Second, the spillover effects of urbanization should be given more attention. Each county within the CUA should focus on modern urban systems and strengthen joint action for air pollution control based on the WHO’s Air Quality Guidelines and China’s Air Pollution Prevention and Control Law. The sharing of pollution monitoring and control should be further strengthened and developing a pollution emission list should be considered. Regions should strengthen cooperation towards clean transportation, optimized urban spatial layout, environmental sustainability, industrial waste remediation and pollution reduction in order to build a circular and sustainable “blue sky urban agglomeration”.

There are still some shortcomings in this study. First, the driving effect of multi-dimensional urbanization has been explored in this study, but the objective demand of urban planning for the urbanization speed and the negative impact of air pollution on urbanization were not considered. Future studies should make more efforts to explore how to achieve the unity of urban air quality improvement and the healthy development of urbanization under the dual goals of urban development and residential happiness. Second, the spatial regression model considers the influence of spatial dependence on the driving effect. However, it does not detect the driving factors of the PM_2.5_ concentrations of a single research unit and the spillover effects on specific areas. In the future, spatial regression models and geographic weighted regression can be combined to better analyze the driving effect of urbanization on PM_2.5_ concentrations at different levels and analyze the coordinated development of urbanization and air quality from a global and local perspective.

Despite these shortcomings, this research has strong practical and application value. This is the first time a practical analysis of the direct and spillover effects of urbanization on PM_2.5_ concentrations has been carried out in CUA using a multi-dimensional urbanization perspective. The research on the driving impact of urbanization on PM_2.5_ can also be extended to other similar emerging urban agglomerations in China to further explore the coordinated relationship between urbanization and air quality in urban agglomerations. Commercial trade, building development, agricultural operations and other natural indicators can also be used as control variables in driving mechanism research of urban activities on carbon emissions and other related fields.

## 6. Conclusions

In this study, control variables such as the wind speed, relative humidity, temperature, air pressure, commercial trade and building development intensity were included in the regression system and the spatial regression analysis of PM_2.5_ was carried out from the grid and administrative levels. It aims to reveal the difference between the direct and indirect effects of the multi-dimensional urbanization of the CUA on PM_2.5_ concentrations in the context of rapid urbanization. The highlights of the results are as follows. The average annual decline of PM_2.5_ concentrations in the CUA reached 4.6%, but the concentration in 91% of the regions still exceeded 35 μg/m^3^ in 2015. The spatial distribution pattern of “high in the southwest and low in the southeast” was gradually formed. The spatial autocorrelation of PM_2.5_ concentrations was further elaborated and the PM_2.5_ concentrations have a significant positive spatial dependence, forming a spatial dependence pattern of “cold spots in the east and hot spots in the west”. Analysis of influence differences showed that urbanization was not the only factor intensifying PM_2.5_ pollution; commercial trade, the average air pressure and building development intensity were found to have a significant impact on air quality. The direct and spillover effects of multi-dimensional urbanization on PM_2.5_ concentrations were significantly different. The driving effect of land urbanization was found to be stronger than economic and population urbanization, particularly the spillover effects on PM_2.5_ concentrations in neighboring areas.

## Figures and Tables

**Figure 1 ijerph-18-10609-f001:**
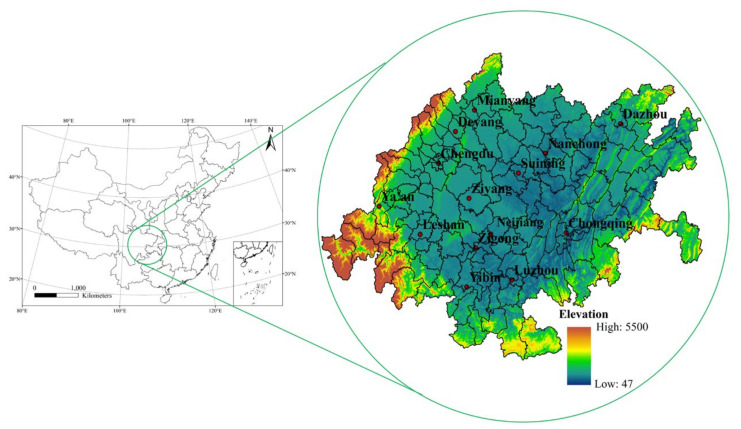
Study area.

**Figure 2 ijerph-18-10609-f002:**
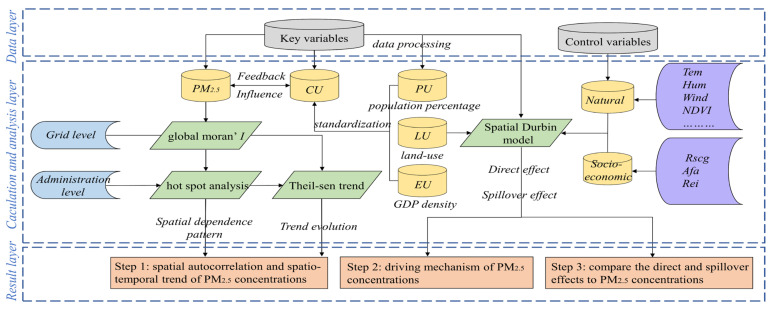
Technical framework of this research.

**Figure 3 ijerph-18-10609-f003:**
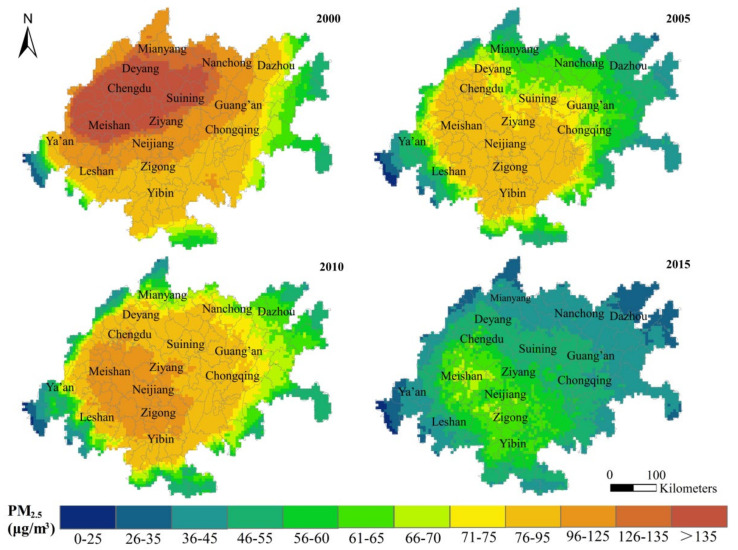
Spatial distribution of PM_2.5_ concentrations in CUA from 2000 to 2015: spatial interpolation was achieved using the natural break point classification of ArcGIS.

**Figure 4 ijerph-18-10609-f004:**
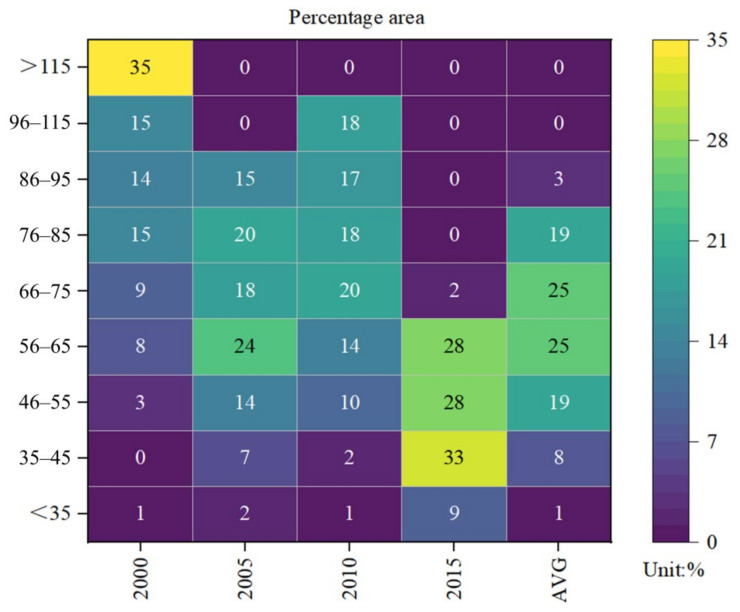
Classification of area proportion of PM_2.5_ concentrations classification in CUA from 2000 to 2015. The results are based on the calculation of the percentage of the grid. AVG was obtained after averaging the PM_2.5_ concentrations over the study period.

**Figure 5 ijerph-18-10609-f005:**
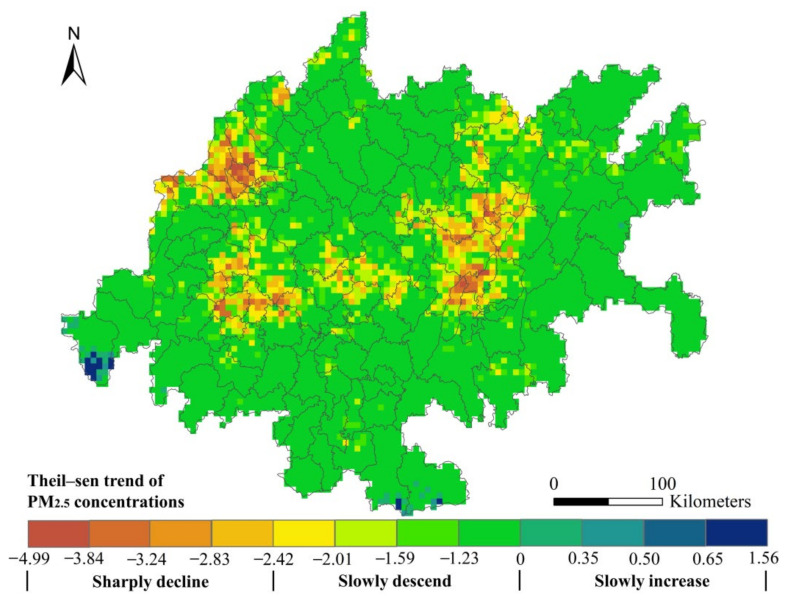
Evolution trend of Theil–Sen index of PM_2.5_ concentrations in CUA from 2000 to 2015.

**Figure 6 ijerph-18-10609-f006:**
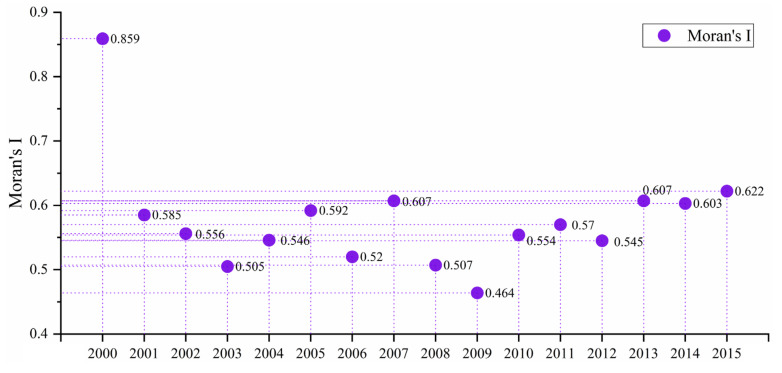
Moran’s *I* of global autocorrelation from 2000 to 2015.

**Figure 7 ijerph-18-10609-f007:**
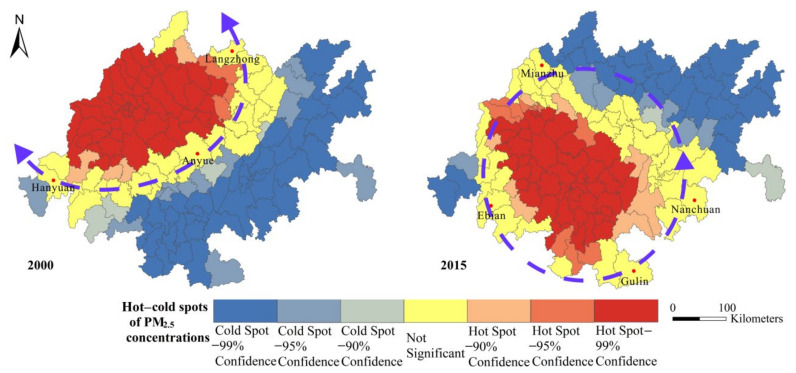
Distribution of cold and hot spots of PM_2.5_ concentrations in CUA in 2000 and 2015.

**Figure 8 ijerph-18-10609-f008:**
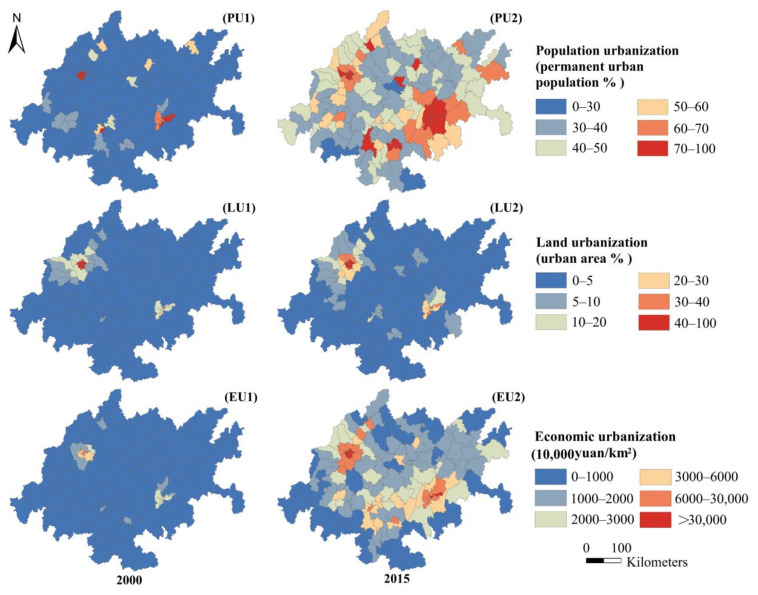
The spatial distribution of multi-dimensional urbanization.

**Table 1 ijerph-18-10609-t001:** Variable category, abbreviation and unit.

Variable Category	Variable	Abbreviation	Measurement Unit
Dependent variable	PM_2.5_ concentrations	*PM_2.5_*	μg/m^3^
Urbanization (key explanatory variable)	Population urbanization	*pop*	people/km^2^
	Land urbanization	*land*	%
	Economic urbanization	*gdp*	10,000 yuan/km^2^
Control variable	Elevation	*dem*	m
	Slope	*slo*	°
	Average annual temperature	*tem*	°C
	Average relative humidity	*hum*	%RH
	Average air pressure	*ap*	Pa
	Average wind speed	*wind*	m/s
	Normalized Difference Vegetation Index	*NDVI*	-
	Average annual precipitation	*pre*	mm
	Per capita retail sales of consumer goods	*rscg*	yuan
	Agricultural fertilizer application	*afa*	t
	Per capita real estate investment	*rei*	yuan

**Table 2 ijerph-18-10609-t002:** SDM estimates of population and comprehensive urbanization of CUA for 2000 and 2015.

Variables	SDM_2000	SDM_2015	Variables	SDM_2000	SDM_2015
*lnpop*	0.012 **	0.028 *	*lncu*	0.078 ***	0.127 **
*lndem*	−0.140 *	−0.624 **	*lndem*	−0.038 *	−0.640 **
*lnslo*	0.075	−0.107 **	*lnslo*	0.066	−0.099 *
*lnap*	0.797 ***	0.495 **	*lnap*	0.830 ***	0.498 ***
*lnwind*	−0.012 **	0.317	*lnwind*	−0.018 *	0.330
*lnndvi*	−0.031 **	−0.050 ***	*lnndvi*	−0.079 **	−0.097 *
*lnrscg*	0.018 **	0.114 **	*lnrscg*	0.032 ***	0.157 ***
*lnrei*	0.207 ***	−0.022	*lnrei*	−0.001	−0.021
*W*lnpop*	0.018 **	0.019*	*W*lncu*	0.202***	0.127*
*W*lndem*	0.173 *	0.579 *	*W*lndem*	0.049	0.587 *
*W*lnslo*	−0.139 **	−0.210 **	*W*lnslo*	−0.095 **	−0.223**
*W*lnap*	−0.861 ***	−0.562	*W*lnap*	−0.896 ***	−0.571 **
*W*lnwind*	−0.033 *	−0.113 **	*W*lnwind*	−0.032 *	−0.121 **
*W*lnndvi*	0.105 *	0.232 **	*W*lnndvi*	0.166 **	0.285 ***
*W*lnrscg*	0.123 *	−0.027	*W*lnrscg*	0.205 **	−0.038
*W*lnrei*	0.243 ***	0.096**	*W*lnrei*	0.107 **	0.077 **
R2	0.756	0.948	R2	0.853	0.951
log-likelihood	313.256	319.107	log-likelihood	304.427	320.664
LR-SLM	40.132 ***	60.236 ***	LR-SLM	10.256 ***	27.892 ***
LR-SEM	32.195 ***	45.371 ***	LR-SEM	9.681 ***	15.297 ***

Note: *cu* indicates comprehensive urbanization, which was obtained as follows: first, the mean standardized economic, land and population urbanization was calculated; second, the sum value was standardized to obtain the final comprehensive urbanization. *, ** and *** indicate the significance at the confidence level of 10%, 5% and 1%, separately.

**Table 3 ijerph-18-10609-t003:** The impact of multi-dimensional urbanization on PM_2.5_ concentration in 2000 and 2015.

Row Number	Type of Urbanization	Variables	Indicator Abbreviation	Direct Effect		Spillover Effect	
				2000	2015	2000	2015
1	Population urbanization	Percentage of permanent urban population	*lnpop*	0.003 (0.078)	0.083 *** (2.685)	2.266 * (4.871)	1.370 * (2.545)
2	land urbanization	Ratio of urban land area	*lnland*	0.220 *** (1.990)	0.209 *** (3.447)	10.785 ** (2.240)	3.717 * (2.761)
3	economic urbanization	GDP density	*lngdp*	0.179 * (1.757)	0.160 ** (2.120)	6.915 ** (1.823)	6.337 * (1.879)
4	Comprehensive urbanization	Based on PU, LU and EU	*lncu*	0.169 *** (2.820)	0.155 *** (2.969)	7.328 * (1.683)	2.151 ** (2.318)

Notes: t-statistics in parentheses. *, ** and *** indicate the significance at the confidence level of 10%, 5% and 1%, separately.

## Data Availability

The data used to support the findings of this study are available from the corresponding author upon request.

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
