# Peer review of "The Direct and Spillover Effect of Multi-Dimensional Urbanization on PM2.5 Concentrations: A Case Study from the Chengdu-Chongqing Urban Agglomeration in China"

_ijerph, 2021, doi:10.3390/ijerph182010609_

Round 1

Reviewer 1 Report

Direct and spillover effect of multi-dimensional urbanization on PM2.5 concentrations: A case study from the Chengdu-3 Chongqing urban agglomeration in China

This is a thorough analysis of what can possibly influence the PM2.5 concentration in an urban agglomerate in China. The manuscript is clearly written with a more or less good chapter layout. But there are things that need more attention.

The manuscript is sent to a journal where the readers are not specialized in the subject of the study. It would therefore be beneficial if some expressions and technical terms are explained as they are introduced in the text. Things like spillover effects, hot and cold spots, etc.

P 1, l 41: “Long-term PM2.5 endangers large” To short and thus incorrect. I suggest something like “Long-term exposure to high concentrations of PM2.5 endangers large”

Figure 1: Caption text is too small. Does not at all explain what is seen in the figure.

Table 1: Same as Figure 1. Better caption needed.

Table 1: “Person” misspelled as is “urbanization” for land.

P 4, l 126: The text gives the impression that the reader should see 142 sample units in figure 1. There are no 142 units to see in Figure 1.

P 4, l 126-128: The sentence is a repetition of what is already explained.

Figure 2. In the middle cell “standardization” is written on top of a line and difficult to see. The figure caption can maybe be more explanatory.

Throughout the manuscript. Please check “Moran’s”. The “s” is missing in a few places.

P 6, l 209: Isn’t a “mean” sign missing over the x in denominator?

P 6. L 217: What does “d” stand for?

P 6, l 284-288: A reference to WHO is missing and this makes it hard to interpret what the authors mean by “threshold for air pollution”. WHO is not publishing thresholds for pollutants. WHO publish air quality guidelines (AQG). Since 2005 the AQG for annual concentrations of PM2.5 is 10 µg/m3. This changed on September 22 2021, when the annual AQG for PM2.5 was lowered to 5 µg/m3 in a new publication from WHO: WHO global air quality guidelines. Please clarify these sentences, and if needed how it influences the outcomes of the study.

Figure 4 and throughout the manuscript. I oppose the use of so many significant digits in the %-numbers presented. The underlaying data and models rely on many assumptions, databases for data are prone to contain assumptions and estimations of presented data. With that as a background I do not believe the authors can really present their own findings in for example PM2.5 concentration classifications in Figure 4 with two decimals (i.e. four significant digits) for these values. Please round to two significant digits, which in most cases means no decimals.

Figure 4: How is AVG calculated? There are four numbers presented (for 2000, 2005, 2010, 2015) and taking the average of these numbers does not give the AVG value. The AVG is not used in the text, as far as I can see.

P 7, l 290: This text is connected to Figure 4 but it is not clear what “value” is referring to. The only “value” given in the sentence is 90.55% but nothing like this is seen in Figure 4. Clarification needed.

P 8-9: The terminology is the same in the text and in the Figure. For example, “slowly increasing” in the text is “slowly rising” in the Figure.

Figure 6: A little more explanatory text would be good for those readers who go directly to the figures to see if the text can be interesting to read.

Figure 7: Here there are no decimals for the %, which is more reasonable.

P 11, l 376-377: The accuracy given for the numbers for these variables is not possible to achieve in this study.

Table 3: Please add the “*, **” explanation from Table 2 to this table.

P 16, l 562-570: Isn’t this text a conclusion? In this case it should be moved.

P 16, Conclusions: First paragraph is explaining the details of the study and there are no conclusions here. The second part gives highlights. This looks more like conclusions, so why not call it conclusions? In the introduction an aim was presented. The conclusions should tell the readers what can be concluded from the study to reach this aim.

Author Response

Direct and spillover effect of multi-dimensional urbanization on PM2.5 concentrations: A case study from the Chengdu-3 Chongqing urban agglomeration in China

This is a thorough analysis of what can possibly influence the PM2.5 concentration in an urban agglomerate in China. The manuscript is clearly written with a more or less good chapter layout. But there are things that need more attention.

Thank you for your many valuable suggestions, these suggestions are of great value for us to improve the quality of the manuscript. Thank you again.

1.The manuscript is sent to a journal where the readers are not specialized in the subject of the study. It would therefore be beneficial if some expressions and technical terms are explained as they are introduced in the text. Things like spillover effects, hot and cold spots, etc.

Response: Thank you for your suggestions. We agree with your point of view. In the article, we have added explanations of spillover effects, hot spots, and cold spots to enhance readers' understanding. For details, see line:52-53;222-225;245-249.

2.P 1, l 41: “Long-term PM2.5 endangers large” To short and thus incorrect. I suggest something like “Long-term exposure to high concentrations of PM2.5 endangers large”

Response: Yes, you are right. The statement here is wrong. We have modified the original sentence according to your suggestion and replaced it with “Long-term exposure to PM2.5 increases the survival risk of residents and causes nearly 1.3 million premature deaths in China every year”. For details, see line: 41-43.

3.Figure 1: Caption text is too small. Does not at all explain what is seen in the figure.

Response: Thank you for your suggestion. According to your suggestion, we have modified the font size in Figure 1, see Figure 1 for details. Thank you again for your suggestion.

4.Table 1: Same as Figure 1. Better caption needed.

Response: Thank you for your suggestion. The font size of the form and the main text are both 10 point. At the same time, we have modified the title of the Table 1 according to your suggestion. See Table 1 for details.

5.Table 1: “Person” misspelled as is “urbanization” for land.

Response: Thanks for your reminder, we rechecked the spelling of Table 1. We need to explain for you is that the abbreviation for population urbanization is pop, and the unit is peason/km2, and the abbreviation for land urbanization is land, and the unit is %.

6.P 4, l 126: The text gives the impression that the reader should see 142 sample units in figure 1. There are no 142 units to see in Figure 1.

Response: Thank you for your suggestions. After your reminder, we have modified Figure 1. See Figure 1 for details. Thank you again for your suggestions.

7.P 4, l 126-128: The sentence is a repetition of what is already explained.

Response: Thank you for your suggestion. We have deleted the original sentence based on your suggestion and amended it to “Considering data availability, the study period was set to 2000-2015, and all spatial data projections were set to Krasovsky 1940 Albers.” See line:127-129 for details.

8.Figure 2. In the middle cell “standardization” is written on top of a line and difficult to see. The figure caption can maybe be more explanatory.

Response: Thank you for your suggestion. We adjusted the position of the text in Figure 2 according to your suggestion, so that readers can see the content of the cell more clearly. See Figure 2 for details.

9.Throughout the manuscript. Please check “Moran’s”. The “s” is missing in a few places.

Response: Thank you for your suggestion. We checked the spelling of “Moran’s” in the manuscript according to your reminder and corrected the content. For details, see line: 210.

10.P 6, l 209: Isn’t a “mean” sign missing over the x in denominator?

Response: Thank you for your suggestion. Here is an omission. After your reminder, we added the “mean” sign. Thank you again for your reminder. See line: 211 for details.

11.P 6. L 217: What does “d” stand for?

Response: What needs to be explained to you is that Gi*(d) is the abbreviation of Getis-OrdGi* index, where d stands for index.

12.P 6, l 284-288: A reference to WHO is missing and this makes it hard to interpret what the authors mean by “threshold for air pollution”. WHO is not publishing thresholds for pollutants. WHO publish air quality guidelines (AQG). Since 2005 the AQG for annual concentrations of PM2.5 is 10 µg/m3. This changed on September 22 2021, when the annual AQG for PM2.5 was lowered to 5 µg/m3 in a new publication from WHO: WHO global air quality guidelines. Please clarify these sentences, and if needed how it influences the outcomes of the study.

Response: Thank you for your suggestion. You are correct. In 2021, AQG has updated the annual average PM2.5 target value to 5µg/m3 based on the new evidence that low-level concentration is beneficial to health. However, since the study period of this article is 2000-2015, we did not refer to the latest AQG of the 2021 version. Your suggestion is very scientific and reasonable. We marked the version of AQG as 2005 in the article and deleted the expression of air pollution threshold. See line: 288-290 for details.

13.Figure 4 and throughout the manuscript. I oppose the use of so many significant digits in the %-numbers presented. The underlaying data and models rely on many assumptions, databases for data are prone to contain assumptions and estimations of presented data. With that as a background I do not believe the authors can really present their own findings in for example PM2.5 concentration classifications in Figure 4 with two decimals (i.e. four significant digits) for these values. Please round to two significant digits, which in most cases means no decimals.

Response: Thank you for your suggestions. We believe that your suggestions and considerations are scientific and reasonable. We have modified the use of decimals in Figure 4 according to your suggestions. See Figure 4 for details.

14.Figure 4: How is AVG calculated? There are four numbers presented (for 2000, 2005, 2010, 2015) and taking the average of these numbers does not give the AVG value. The AVG is not used in the text, as far as I can see.

Response: Thank you for your suggestion. AVG is the classification result of the average PM2.5 concentration of Chengdu-Chongqing urban agglomeration obtained after averaging the PM2.5 concentration from 2000 to 2015 (ie., 16 years), although we only show the calculation results for 4 years in Figure 4. We have modified AVG’s interpretation based on your suggestions, see line: 300-301 for details. In addition, although the AVG is not specifically described in the article, the calculation of the AVG value can help readers understand that the PM2.5 concentration of the Chengdu-Chongqing urban agglomeration still has a significant pollution risk throughout the period, so we did not exclude it. Thanks again for your valuable suggestions.

15.P 7, l 290: This text is connected to Figure 4 but it is not clear what “value” is referring to. The only “value” given in the sentence is 90.55% but nothing like this is seen in Figure 4. Clarification needed.

Response: Thank you for your suggestion. “Value” refers to the proportion of the area with a concentration of more than 35μg/m3 in the Chengdu-Chongqing urban agglomeration. We have modified the original sentence based on your suggestion. See line: 291-294 for details.

16.P 8-9: The terminology is the same in the text and in the Figure. For example, “slowly increasing” in the text is “slowly rising” in the Figure.

Response: Thank you for your suggestions. We have unified the contents of the two places according to your suggestions. See Figure 5 for details.

17.Figure 6: A little more explanatory text would be good for those readers who go directly to the figures to see if the text can be interesting to read.

Response: Thank you for your suggestion. Your suggestion is very good, but we generally show the changes of Moran’s I in this form. We do not have a better alternative for the time being, so we did not modify the picture.

18.Figure 7: Here there are no decimals for the %, which is more reasonable.

Response: Thank you for your affirmation. The legend here is the fixed output format of ArcGIS.

19.P 11, l 376-377: The accuracy given for the numbers for these variables is not possible to achieve in this study.

Response: Yes, we agree with your point of view. The results of the regression model must have a certain degree of error, but after careful consideration and re-checking, we found that the results obtained are normal. The large value may be related to the small coefficient itself, and this phenomenon does not affect our judgment of the influence of urbanization.

20.Table 3: Please add the “*, **” explanation from Table 2 to this table.

Response: Thank you for your suggestion. We have added an explanation after Table 3 based on your suggestion, see Table 3 for details.

21.P 16, l 562-570: Isn’t this text a conclusion? In this case it should be moved.

Response: Thank you for your suggestion. What needs to be explained to you is that this part of the content refers to the applicability of section 5.3. It is mainly used to illustrate the study value and corresponds to the shortcomings, so we do not plan to move it to maintain this structure. Thank you again for your suggestions.

22.P 16, Conclusions: First paragraph is explaining the details of the study and there are no conclusions here. The second part gives highlights. This looks more like conclusions, so why not call it conclusions? In the introduction an aim was presented. The conclusions should tell the readers what can be concluded from the study to reach this aim.

Response: Thank you for your suggestion. We agree with your point of view. Our original intention is to introduce the innovation of this research compared with previous research in the first paragraph. According to your suggestion, we have adjusted the content of the first paragraph, see line: 576-580 for details. In addition, according to your suggestions, we have added an statement of the study goals achieved in the conclusion, see line: 585-592 for details.

Reviewer 2 Report

The manuscript presents a study of factors impacting the PM2.5 concentrations at Chengdu-Chongqing urban agglomeration, China. The manuscript is well written and can be published after a minor revision.

Specific comments:

  1. In caption of Figure 1should specify what the labels on the right represent. Also, the labels were hard to read.
  2. Section 2.2 needs more explanation as to how this data is derived. Does it relate in any way to the 142 sample units mentioned on line 126?
  3. Lines 185-187: “For example, Wang et al. (2017) concluded that international trade affects the global distribution of air pollution and that air pollutant emissions are related to the goods and services consumed.” This needs further explanation, as to how international trade affects air pollution.
  4. Figure 3 – It should be mentioned in the caption, as to how was the spatial interpolation done.
  5. Lines 288-291: “In 2000, more than 99.44% of the CUA had concentrations of more than 35μg/m3. In 2015, while the proportion decreased to 90.55%, the value was still very high (Figure 4), indicating that air pollution was still a major problem in this ..”. What is presented in Figure 4 needs to be explained a bit better. Is it the percentage of time that 35μg/m3 was exceeded? It also needs to be specified in the caption of Figure 4.
  6. Line 451: “adapte” was it meant to be “adapted”?

Author Response

The manuscript presents a study of factors impacting the PM2.5 concentrations at Chengdu-Chongqing urban agglomeration, China. The manuscript is well written and can be published after a minor revision.

Thank you for your affirmation of the manuscript. We are greatly encouraged. Your suggestions are also of great value in improving the quality of the manuscript. Thank you again for your suggestions.
Specific comments:

1.In caption of Figure 1should specify what the labels on the right represent. Also, the labels were hard to read.

Response: Thank you for your reminder. The label on the right means Elevation. We adjusted the size of the text in the Figure 1 according to your reminder. See Figure 1 for details.

2.Section 2.2 needs more explanation as to how this data is derived. Does it relate in any way to the 142 sample units mentioned on line 126?

Response: Thank you for your suggestions. We summarized the extraction of all spatialized data in section 2.1 based on your suggestions. We explain to you: all spatial data projections were extracted into 142 administrative regions based on Zonal Statistics tool of ArcGIS. See line: 127-129 for details.

3.Lines 185-187: “For example, Wang et al. (2017) concluded that international trade affects the global distribution of air pollution and that air pollutant emissions are related to the goods and services consumed.” This needs further explanation, as to how international trade affects air pollution.

Response: Thank you for your suggestion. This study mainly evaluates the air quality burden brought by commodity trade from two aspects: international exports and inter-provincial trade. They believe that the production of goods often requires inputs of raw materials and energy forms produced by heavy industry, which easily leads to the cross-regional transmission of pollutant components. Based on your suggestion, we have revised the expression of this sentence, see line: 186-189 for details.

4.Figure 3 – It should be mentioned in the caption, as to how was the spatial interpolation done.

Response: Thank you for your reminder. We explain to you that the spatial interpolation here is to classify the PM2.5 remote sensing image by natural breakpoints and then realize the mapping. We explained in the title according to your suggestion, see Figure 3 for details.

Lines 288-291: “In 2000, more than 99.44% of the CUA had concentrations of more than 35μg/m3. In 2015, while the proportion decreased to 90.55%, the value was still very high (Figure 4), indicating that air pollution was still a major problem in this ..”. What is presented in Figure 4 needs to be explained a bit better. Is it the percentage of time that 35μg/m3 was exceeded? It also needs to be specified in the caption of Figure.

Response: Thank you for your suggestion. Based on your suggestion, we have modified the content of this sentence to better reflect the Figure 4 and better read. For details, see line: 291-294; 299-300. Thanks again for your suggestions.

6.Line 451: “adapte” was it meant to be “adapted”?

Response: Yes, you are correct. Here is a spelling error. We have corrected the content according to your suggestions. See line: 456 for details.